# In Pursuit of Genetic Prognostic Factors and Treatment Approaches in Secondary Acute Myeloid Leukemia—A Narrative Review of Current Knowledge

**DOI:** 10.3390/jcm11154283

**Published:** 2022-07-23

**Authors:** Paulina Stefaniuk, Agnieszka Szymczyk, Monika Podhorecka

**Affiliations:** 1Department of Hematooncology and Bone Marrow Transplantation, Medical University of Lublin, Staszica 11 Street, 20-080 Lublin, Poland; monika.podhorecka@onet.pl; 2Doctoral School, Medical University of Lublin, Witolda Chodźki 7 Street, 20-059 Lublin, Poland; 3Department of Clinical Transplantology, Medical University of Lublin, Chodźki 7 Street, 20-093 Lublin, Poland; agusia_szymczyk@wp.pl

**Keywords:** acute myeloid leukemia, secondary AML, therapy-related AML, myelodysplastic syndromes, genetic prognostic factors

## Abstract

Secondary acute myeloid leukemia can be divided into two categories: AML evolving from the antecedent hematological condition (AHD-AML) and therapy related AML (t-AML). AHD-AML can evolve from hematological conditions such as myelodysplastic syndromes, myeloproliferative neoplasms, MDS/MPN overlap syndromes, Fanconi anemia, and aplastic anemia. Leukemic transformation occurs as a consequence of the clonal evolution—a process of the acquisition of mutations in clones, while previous mutations are also passed on, leading to somatic mutations accumulation. Compared de novo AML, secondary AML is generally associated with poorer response to chemotherapy and poorer prognosis. The therapeutic options for patients with s-AML have been confirmed to be limited, as s-AML has often been analyzed either both with de novo AML or completely excluded from clinical trials. The treatment of s-AML was not in any way different than de novo AML, until, that is, the introduction of CPX-351—liposomal daunorubicin and cytarabine. CPX-351 significantly improved the overall survival and progression free survival in elderly patients with s-AML. The only definitive treatment in s-AML at this time is allogeneic hematopoietic cell transplantation. A better understanding of the genetics and epigenetics of s-AML would allow us to determine precise biologic drivers leading to leukogenesis and thus help to apply a targeted treatment, improving prognosis.

## 1. Introduction

Acute myeloid leukemia (AML) is the most common acute type of leukemia in adult patients, estimated to be approximately 80% of all cases of acute leukemias in this group [1]. AML is generally a rare malignancy, with an incidence rate of 3.7 cases/ 100,000 per year in Europe [2]. In Poland, between 2004 and 2010, approximately 520 new incidences were reported in adults annually, with an incidence rate of 1.8 cases/100,000 per year. Patients of more than 70 years old constituted the most numerous group [3]. The incidence in adults rises with age, with the average age at the time of diagnosis reaching 65 years [4]. In patients younger than 65 years old, the incidence is 1.3/100,000 cases, whereas in patients over 65 years old it accounts for 12,2 cases [1]. The five-year overall survival rate is reached in 60–75% children and adolescents [5], compared to only 30% of patients over 65 years old surviving more than one year after diagnosis [1]. The five-year AML-specific survival rate has doubled since the late 1970s in all age groups as a result of continual reassessment of the treatment standards and a wider availability of novel medicines. Additionally, supportive care has been improved, as well as the prophylaxis of bacterial and fungal infections. The outcomes of patients with AML undergoing hematopoietic stem cell transplantation (HSCT) improved in three study time periods (1980–1988, 1989–1997, and 1998–2005) in all age groups [6]. 

According to the World Health Organization (WHO), AML is a group of neoplastic disorders in which at least 20% of cells in the blood or bone marrow are myeloblasts [7]. Since the third edition of WHO Classification of Tumors of Hematopoietic and Lymphoid Tissues, genetic abnormalities have been included in the diagnostic algorithms, which allow us to diagnose AML even without the presence of “the ≥20% criterion”. Those exceptions include AML with cytogenic abnormalities (CBF AML): t(8;21), inv(16), or t(16;16). *NPM1* mutated AML and acute promyelocytic leukemia. The revised fourth edition of the WHO classification classifies AML into six types: AML with recurrent genetic abnormalities; AML with myelodysplasia-related changes (MRC); therapy-related myeloid neoplasms (t-MN); AML, not otherwise specified (NOS); myeloid sarcoma; and myeloid proliferations related to Down syndrome [7]. 

As mentioned before, AML is not a homogenous disease, but a group of biologically and clinically heterogenous bone marrow malignancies. AML can arise de novo (primary AML) or as a consequence of a prior malignancy (secondary AML) [8]. According to the definition, patients with primary AML have no history of chronic myeloid disorders, myelodysplastic syndrome, myeloproliferative disorders and have not been previously exposed to therapies and agents which can result in leukemogenesis [8]. Secondary AML, which is the main subject of this article, accounts for approximately 25–30% of AML cases [8]. It can be divided into two categories: AML evolving from antecedent hematological condition (AHD-AML) and therapy related AML (t-AML) [9]. 18–20% of all AML cases evolve from a previous hematological disease and 6–8% are therapy related [10,11,12]. Compared to primary AML, secondary AML is generally associated with poorer prognosis [9,13]. Moreover, the treatment regimens and their efficiency in secondary AML differ significantly [14]. As a result of the frequent exclusion of secondary AML patients from clinical trials, knowledge of this AML category remains scarce and needs to be furtherly evaluated. This article summarizes information regarding the epidemiology, pathogenesis, prognostic factors, and treatment approaches in secondary AML.

## 2. AML Evolving from Antecedent Hematological Conditions

Secondary AML can evolve from other hematological conditions, most frequently from myelodysplastic syndromes. All myeloid malignancies can become AML, including myeloproliferative neoplasms [MPN] (primary myelofibrosis, polycythemia vera, essential thrombocytopenia) and MDS/MPN overlap syndromes (chronic myelomonocytic leukemia, atypical chronic myeloid leukemia). Leukemic transformation is also observed in patients with bone marrow failure symptoms such as Fanconi anemia and aplastic anemia [10]. The risk of evolution to AML varies significantly depending on primary myeloid malignancy type [8].

Secondary AML (s-AML) differs biologically from de novo AML. The most important features of s-AML in context of differentiation with de novo AML are the presence of multilineage dysplasia, complex karyotype with genetic material loss in most cases, and pathogenesis connected with sequential acquisition of somatic mutations. Patients with s-AML are thought to harbor more mutated genes than patients with de novo AML. In [15], mutations in *SRSF2*, *SF3B1*, *U2AF1*, *ZRSR2*, *ASXL1*, *EZH2*, *BCOR*, or *STAG2* were >95% specific for the diagnosis of s-AML. Walter et al. [16] performed the whole-genome sequencing of skin and bone marrow obtained during both the MDS and AHD-AML stage, in seven patients with s-AML. AML bone marrow samples were not monoclonal but were in fact a mosaic of genomes with different sets of mutations. In both MDS and s-AML samples, about 85% of bone marrow cells were clonal. In s-AML samples the authors found 11 recurrent mutations in coding genes: *CDH23*, *NPM1*, *PTPN11*, *RUNX1*, *SMC3*, *STAG2*, *TP53*, *U2AF1*, *UMODL1*, *WT1*, and *ZSWIM4.* At least one mutation in a coding gene was observed in patients who progressed to s-AML. Moreover, in all patients who progressed to s-AML, the antecedent founding clone containing 182 to 660 somatic mutations was carried forward, and the outgrowth or emergence of at least one subclone with new mutations occurred. In patients who progressed rapidly, the prevalence of an s-AML specific mutation was lower than in patients with progression lasting more than 20 months.

One of the myeloid conditions that have the propensity to evolve into AML is myelodysplastic syndrome (MDS). It is a heterogeneous group of neoplasms in which hematopoiesis is ineffective and dysplastic, which results in peripheral blood cytopenias [17]. MDS is the most common cause of non-congenital bone marrow failure in adult patients. Lower risk MDS patients have a 5% at 1 year and 10% at 2 years risk of leukemic transformation, whereas in MDS with excessive blasts, the risk is estimated to be 25% after at 1 year and 35% at 2 years [10]. DNA sequencing techniques provided us with insight into the pathogenesis of MDS [18]. 

MDS arises through the sequential acquisition of somatic mutations in a set of recurrently involved genes. To fully (obviously within the limits of current knowledge) understand the pathogenesis of AML antecedent to MDS, one must analyze the process from the very beginning. It seems that the earliest genetic step in MDS pathogenesis is “clonal hematopoiesis of undetermined potential” (CHIP)—a term introduced by Steensma et al. [19]. CHIP describes the situation in which somatic mutations are present in the blood or bone marrow cells and no other criteria for hematologic neoplasia are met. CHIP is a significant and independent prognostic factor for the individuals with higher risk of malignant transformation [18]. Genovese et al. [20] proved that the risk of developing hematological malignancy in patients with CHIP is 12.9 times higher than in healthy individuals. Clonal hematopoiesis, most frequently associated with somatic mutations in genes *DNMT3A*, *ASXL1*, and *TET2*, was observed in 10% of people older than 65 years old and only in 1% of patients younger than 50 years old. Although the presence of CHIP is associated with the higher risk of myeloid malignancies, the absolute risk of malignant transformation is estimated for only 0.5–1% per year [19]. Thus, the identification of factors that promote the transformation from CHIP to s-AML seems useful; it seems that it is not only the matter of CHIP mutations that are useful, but also the germline polymorphism and cell extrinsic factors. During MDS initiation, newly acquired mutations within the previous mutated clone appear, while previous mutations are also passed on. Those acquired mutations may have no consequences as “passenger” mutations or become driver mutations, contributing to clonal evolution [8]. The final stage of the disease progression is an evolution from MDS to s-AML, occurring as a consequence of somatic mutations accumulation [18]. Figure 1 shows the process of clonal evolution schematically.

Shukron et al. [21] suggested an alternative theory: the multiple mutations that characterize s-AML are a consequence of a permissive mutation-selection process, which is preceded by a single gene-driven transformation event. Following this hypothesis, the risk of s-AML is expected to be constant in time. The fact that the majority of patients with MDS present an abrupt type of progression supports this theory. This thesis is based on the authors’ research, conducted by Shukron et al., who analyzed the population of 1079 patients with MDS. In the study, the risk of transformation of MDS to s-AML was constant in time elapsed from MDS diagnosis in IPSS low, intermediate-1, and intermediate-2 risk groups. However, for the IPSS high risk group, the risk of transformation decreased over time. According to the authors’ theory, this result could be attributed to the residual heterogeneity of the IPSS high group, which included patients who in fact already had s-AML, rather than high-risk MDS. After eliminating the patients whose blast percentage was more than 20% from the IPSS high group, the s-AML risk was also constant in the IPSS high group. The number of patients with high-risk cytogenetics increased with the IPSS risk category [21].

To diagnose AML with myelodysplasia related changes (AML-MRC), there should be at least 20% of blasts in blood or bone marrow, the patient should have a history of myelodysplastic syndrome (MDS), MDS/myeloproliferative neoplasm (MPN), or MDS-related cytogenetic abnormality or multilineage dysplasia, and patients should not have received cytotoxic or radiation therapy for an unrelated disease or have the recurrent cytogenetic abnormalities described for AML with recurrent genetic abnormalities [22]. To diagnose AML-MRC based solely on morphology, at least two myeloid lineages have to be present with at least 50% dysplasia. MDS-related cytogenetic abnormalities, according to WHO 2016 classification include: complex karyotype (3 or more abnormalities), −7/del(7q), del(5q)/t(5q), i(17q)/t(17p), −13/del(13q), del(11q), del(12p)/t(12p), idic(X)(q13), t(11;16)(q23.3;p13.3), t(3;21)(q26.2;q22.1), t(1;3)(p36.3;q21.2), t(2;11)(p21;q23.3), t(5;12)(q32;p13.2), t(5;7)(q32;q11.2), t(5;17)(q32;p13.2), t(5;10)(q32;q21.2), t(3;5)(q25.3;q35.1) [7]. Walter et al. [16] are of the opinion that the distinction between MDS and s-AML should rely on identifying pathogenic mutations and their clonality, rather than on counting the percentage of the blast. 

## 3. Mutations Leading to Leukemic Transformation

The driver mutations that may lead to s-AML can be divided into mutations in components of spliceosome, epigenetic regulators, cohesion components, transcription factors and signal transduction molecules [23]. 

Mutated splicing factors, especially *SF3B1*, *SRSF2*, *U2AF1*, and *ZRSR2* are found in about 60% of patients with MDS [18]. RNA splicing is the processes in which pre-mRNA is catalyzed by the spliceosome, whose function it is to coordinate the intron excision and exon ligation during the generation of mature messenger RNA [18,24]. The expression of splicing factors leads to the generation of alternate transcripts, and, in consequence, the oncogenic proteins [25]. *SRSF2* and *U2AF1* are proven to be the factors that increase the risk of leukemic transformation in MDS, with four-fold and three-fold higher risk, respectively [8].

The category of epigenetic regulators includes genes involved in histone modifications and DNA methylation [8,18]. Generally, epigenetics is defined as modifications to the genome that occur without direct connection to the DNA sequence. Impaired epigenetic alterations may cause inappropriate onset of genetic expressions and, hence, lead to cancer development [26]. *ASXL1* and *EZH2* are epigenic regulators involved in chromatin remodeling, localized on chromosome 20 and 7, respectively [8,27]. *EZH2* is recurrently mutated in MDS, PMF, and secondary MF, and, on the other hand, is rare in PV and ET. *ASXL1* mutation is a poor prognostic factor that makes the leukemic transformation more probable [8]. It has also been proven that *ASXL1* frameshift mutations predict a worse outcome after allogeneic HSCT in patients with MDS and s-AML [28]. *EZH2* loss-of-function mutation is present in 5% of patients with MDS [18], 6% of patients with PMF, 4% of patients with AML, and 1–3% of patients with EF and PV [8,18,29]. The majority of mutations in *EZH2* are located in the catalytic SET domain [30]. A loss of *EZH2* activity is associated with poor prognosis in myeloid malignancies [31]. However, in many other malignant neoplasms such as lymphoma, melanoma, and prostate and breast cancer, *EZH2* overexpression, and not loss-of-function, is associated with worse PFS [32]. 

In AML, *EZH2* mutations often co-occurred with *CEBPA*, *ASXL1*, *TET2*, and *RAD21* mutation. Both *EZH2* mutations and low *EZH2* expression were associated with a trend towards an increased risk of death in patients with AML receiving standard chemotherapy [33]. Kempf et al.’s [29] study suggests that the *EZH2* loss-of-function mutation induces resistance against cytarabine in the cell lines HEK293T and K562 and in patient-derived xenograft model. Resistance is attributed to the upregulation of genes involved in apoptosis, proliferation, and transmembrane transport, which results in the cell selective growth advantage [29]. Mutations in *TET2*, *IDH1*, *IDH2*, and *DNMT3α* impair myelopoiesis through aberrant DNA methylation [8]. *TET2* codes for the enzyme that catalyzes the hydroxylates methylated cytosine [18]. Mutations in this gene are present in MDS, MDS/MPN overlap syndromes, MPN, and s-AML with the prevalence of approximately 10–26%, 22–58%, 7–13%, and 24–32%, respectively [34]. The role of *TET2* mutations in myeloid neoplasms is unclear: some studies show them as the important factor for leukemic progression [35], while others do not [36]. *TET2* activity is affected by mutations in mutated isoforms 1 and 2 of isocitrate dehydrogenases (*IDH1* and *IDH2)* [18]. Neoplasms with *IDH1* and *IDH2* mutated share abnormal histone and DNA methylation which cause impaired stem cell differentiation and, hence, tumorigenesis [37]. Mutations in either *IDH1* or *IDH2* are heterozygous missense mutations, affecting approximately 20% of patients with AML [38]; they are present in 12% of all MDS cases. Their role in MDS progression to s-AML remains unclear: some studies report an increased risk of progression to s-AML with *IDH1* mutations only, while other authors insist that the risk of progression is enhanced with either mutation [8]. 

Mutations of transcription factors, including *RUNX1*, *ETV6*, *IKZF1*, *CUX1*, *TP53*, and *PHF6* were proven to be involved in leukemic progression of chronic myeloid malignancies. One of the genes of the greatest interest in this context is *RUNX1* [8]. *RUNX1* mutations are present in MDS, CMM, L and secondary AML with a frequency of 10%, 37%, and 10%, respectively. *RUNX1* transcription factor regulates both normal and abnormal hematopoiesis; the lack of *RUNX1* activity leads to hematopoiesis defects and is embryonic lethal [39]. The majority of *RUNX1* mutations are loss-of-functions missense mutations, deletions, or truncation mutations in the homology domain or in the transactivation domain. In MDS patients who progressed to s-AML, *RUNX1* mutations were present three times more frequently than in patients who did not experience progression [8]. Somatic or germline mutations in *RUNX1* predict chemotherapy resistance and poorer outcome in AML [39]. 

Cohesin is a multiprotein complex composed of SMC1A, SMC3, RAD21, and the adapter proteins STAG1/STAG2. Cohesin plays a role as an effector of sister-chromatid cohesin during the metaphase of mitosis [40]. Mutations in *STAG2*, *SMC3*, and *RAD21* predict poor OS in patients with MDS, MPNs, and AML. Cohesin mutations are more prevalent in patients with *RUNX1*, *Ras*-family oncogenes, *BCOR*, and *ASXL1* mutations [40]. Cohesin mutations occur in over half of patients with DS-AMKL (acute megakaryocytic leukemia associated with Down syndrome) [41]. 

Last, but not least, there are mutations in signaling pathways. Mutations in proliferative genes in tyrosine kinase and RAS pathways are found in patients with impending transformation to s-AML. 40% of MDS patients at the time of transformation share mutations in *NRAS*, *KRAS*, and *FLT3*, which is associated with poor prognosis [8]. 

Some of the mutations became targets for novel treatment agents in AML. Those agents are described in the “targeted therapies” section. Numerous approaches to target actionable mutations in the genes described above are under investigation, including the usage of next-generation *FLT3* inhibitors gilteritinib, crenolanib, and quizartinib [42], or the SF3B1 modulator H3B-8800 [10]. Table 1 presents the mutations proven to be involved in leukemic transformation in MDS.

The most prevalent tool to evaluate the risk of leukemic transformation in everyday clinical practice is the Revised International Prognostic Scoring System (R-IPSS). R-IPSS includes bone marrow cytogenetics data, blast percentage, and the presence of peripheral blood cytopenia. Poor prognosis is associated with at least 3 abnormalities or chromosome 7 anomalies, whereas normal karyotype, -Y alone, del(5q) alone, and del(20q) are associated with good prognosis. R-IPSS divides MDS patients into subgroups, in which the risk of leukemic transformation differs significantly [59]. However, since the introduction of the system in 1997, multiple modifications have been proposed [60]. One of the novel approaches, proposed by Gu et al. [61] in 2021, is the integration of IPSS-R and gene mutations, called “mutation combined with revised international prognostic scoring system” (MIPSS-R). The mutation risk stratification includes several mutated genes associated with poor prognosis and one favorable prognostic mutated gene: *SF3B1*. The statistical analysis proved the superiority of MIPSS-R in separating patients into different prognostic subgroups [61]. 

The clinical course of MDS is not only characterized by the risk of transformation to AML and reduced survival in the majority of the patients, but also by the quality of life (QoL). The assessment of QoL provides information on the patient’ s perspective and perception. The assessment of QoL in MDS has been propagated by many clinicians because it was observed that restrictions in QoL may predict poor treatment response [62]. The QLU-C10D is a questionnaire, developed in order to measure cancer patients’ quality of life and to relate it to survival time and treatment costs. Gamper et al. [63] proved that the EORTC QLU-C10D may be useful to determine cancer-specific health state utility values in MDS patients [63]. 

## 4. Leukemic Transformation in Atypical Chronic Myeloid Leukemia

One of the diseases with an especially high rate of leukemic transformation is atypical chronic myeloid leukemia (aCML). aCML is a *BCR-ABL1* negative malignancy with features of both myelodysplastic syndrome and myeloproliferative neoplasm. It is a rare maligancy mainly of the elderly. aCML is characterized by the presence of neutrophilic leukocytosis and the presence of dysgranulopoiesis [64]. 

The prognosis in aCML is very poor, with median OS ranging between 10.8–25 months. 40% of patients transform into AML. Median AML-free survival is 11.2 months [65]. Leukocytes count at presentation over 50 × 10⁹/L, increased immature precursors in the peripheral blood, female gender and an age greater than 65 years, which are poor prognostic factors for OS. Mutations in *ASXL1*, *SETBP1* and *TET2* predict more aggressive disease. Patients with palpable hepatomegaly or splenomegaly, monocytosis, over 5% blasts in bone marrow, marked dyserythropoiesis and transfusional requirement seem to have a higher risk of progression to s-AML [64]. 

## 5. Therapy-Related AML

Another type of s-AML is therapy-related AML (t-AML), defined as an AML in patients previously exposed to chemotherapy or radiation therapy [9]. The Revised 4th edition of the WHO classification uses the term t-MN (therapy-related myeloid neoplasia), rather than t-AML. The T-MN category includes acute and chronic myeloid disorders such as MDS, AML and MDS/MPN overlap syndromes that occur as a complication of chemotherapeutic agents or radiotherapy exposure [8]. 

T-AML is classified into two types that differ in the following ways: from the time of leukemogenic therapy exposure to the AML outset; type of leukemogenic treatment and genetic alternations. Type 1 t-AML occurs 4–7 years after exposure to alkylating chemotherapy or radiotherapy. In two thirds of the cases, it is preceded by MDS and in one third MDS-related changes are observed. The most common genetic alterations in type 1 are unbalanced loss of genetic material of chromosome 5 and/or 7 and *TP53* mutation. Type 2 t-AML occurs 2–3 years after the treatment with topoisomerase inhibitors type II. Type 2 t-AML is not preceded by the MDS phase. Type 2 t-AML patients are more likely to have balanced chromosomal aberrations involving 11q23 (*MLL)* and 21q22 *(RUNX1)* [10]. Survivors of breast cancer and lymphomas constitute the group with the largest number of patients as far as t-AML patients are concerned [8]. As a consequence of increases in cancer survivorship and an increasing number of patients undergoing chemotherapy, including polychemotherapy, the incidence of t-AML is expected to rise [8,66]. 

The most commonly mutated gene in t-AML is *TP53. TP53* is located on chromosome 17 and encodes a tumor suppression protein p53, widely known as the “guardian of the genome” [67]. Activation of p53 takes place in response to DNA damage and replication stress. p53 activation leads to the elimination or repair of damaged cells, in order to reduce the risk of propagating mutations [68]. In approximately 60% of malignant neoplasms in humans, *TP53* is mutated or inactivated [58,69]. The presence of *TP53* mutation is also shown to be one of the most important factors of poor prognosis in t-AML. *TP53* mutation is more frequently observed in t-AML than in de novo AML (16% vs. 8%, respectively) [70]. *TP53* mutations are present not only before the onset of the t-AML, but also before chemotherapy exposure. Hence, it seems that the host genetic susceptibility is a key factor predicting which patients are at increased risk of developing t-AML. It is hypothesized that *TP53* promotes a selective growth advantage after exposure to chemo- or radiotherapy, leading to out-competing cells with high *p53* activity by those with reduced *p53* activity.

Recently, *PPM1D* gain-of-function mutation is also thought to contribute to selective growth advantage upon the chemotherapy exposure, especially in non-*TP53*-mutated t-AML [8]. *PPM1D* is a DNA-damage response regulator, a member of the PP2C family of protein phosphatases. In response to environmental stresses, p53 induces the expression of *PPM1D*, which negatively regulates the cell stress response pathways, resulting in suppression of p53-mediated transcription and apoptosis [58]. *PPM1D* is mutated in 20% of patients with therapy-related myeloid neoplasms [8]. 

The role of inherited cancer susceptibility mentioned before has been proved in a few studies, including Churpek et al.’s [66] study, referring to the genetic landscape of t-AML, following breast cancer. 88 breast cancer survivors were included in the study. 81 women developed t-AML and 7 women had t-ALL. The time from diagnosis of breast cancer to the first bone marrow examination revealing acute leukemia ranged from 28–105 months; median time was 58 months. 78% of patients underwent chemotherapy, most commonly with regimens involving doxorubicin and cyclophosphamide. In 79% of patients, radiation exposure was reported. In 92% of patients, clonal cytogenetic abnormalities were found: most commonly abnormalities of chromosome 5 and/or 7 (in 43 patients) and recurring balanced translocations (in 29 patients). t(9;11)(p22;q23) was the most frequent balanced translocation. The somatic mutations were distributed among 17 genes, the most common being *FLT3* and *TET2.* Some cases of t-AML may not be etiologically related to previous treatment, but are independent secondary primary cancers, for which pathogenesis is related to inherited mutations. 21% of patients who developed therapy-related leukemia had inherited mutations in *BRCA1* (6%), *TP53* (6%) and *BRCA2* (4%), *CHEK2* (2%) and *PALB2* (2%). 60% of patients with inherited mutations had a family history of cancer. This observation highlights the importance of such a simple diagnostic tool as a patient’s medical history in cancer management and prognostication. The authors support the recommendation of genetic testing of all women who develop s-AML with prior breast cancer, in order to implement primary prevention in their close relatives and patients who survive acute leukemia [66].

Another particularly interesting study aiming to evaluate the differences of a t-AML secondary to breast cancer and to lymphoma (Hodgkin’s lymphoma and non-Hodgkin’s lymphomas) is worth mentioning [71]. The median age and time to AML onset were similar in the breast cancer and lymphoma survivors and accounted for 63–64 years and 5 years, respectively. In both groups, the most common treatment included chemotherapy and radiotherapy. The genetic profiles in patients with previous breast cancer were more similar to the genetic profiles in novo AMLs—the majority of patients in this group had the normal karyotype (50%) or recurrent translocations. In contrast, patients with previous lymphoma had a higher proportion of MDS-related cytogenetics, with only 30% of patients having a normal karyotype. Also, the response to chemotherapy differed: CR was achieved in 75% of patients in the breast cancer group, and only 48% in a lymphoma group. This data highlights that t-AML is in fact a heterogeneous group that deserves to be further evaluated in clinical trials and approached in a personalized manner [71].

Hasserjian et al. [72] propose that some putative relapsed AML cases in which cytogenetics genetic profiles are completely unrelated to the original AML should also be referred to as therapy-related AML (or a “new AML”) in order to mark the difference between relapsed disease that shares genetic features with the original AML and s-AML cases that occur most commonly 2–3 years from the “first” AML diagnosis. The authors are of the opinion that, unlike in relapsed disease (defined as the identification at least 5% blasts in the bone marrow after reaching the complete remission), t-AML should be diagnosed only if there are at least 20% blasts in the blood/bone marrow or if the cytogenetic abnormalities that characterize AML are present [72]. 

## 6. Prognosis in Secondary AML

The prognosis in s-AML differs significantly compared to the prognosis in de novo AML. Sheehyun et al. [9] conducted a retrospective study on Korean AML patients, including 437 de novo AML cases, 41 t-AML cases and 66 AHD-AML patients. Patients were divided into cytogenetics group categories. The poor cytogenetic features were significantly more often observed in the AHD-AML group (de novo AML vs. t-AML, *p* = 0.179; de novo AML vs. AHD-AML, *p* < 0.001; t-AML vs. AHD-AML, *p* = 0.321). 30% of patients with t-AML had a prior history of lymphoma and 17% of patients had history of breast cancer. The median time from the primary disease diagnosis and AML onset was 47 months. In the AHD-AML group, as many as 62% of patients had MDS, whereas 27% were diagnosed with MPN. No difference in frequency of poor cytogenetics was observed in patients with MDS-AML and MPN-AML. During intensive chemotherapy, the treatment related mortality was the highest in the AHD-AML group, accounting for 54.5% of patients. Also, AHD-AML was associated with worst overall survival (compared not only with de novo AML, but also with t-AML), irrespective of the cytogenetic risk group and age [9].

Wang et al. [14] compared the genetic alternation patterns and prognosis in s-AML and AML in elderly patients (e-AML) and in patients with de novo AML. The significant advantage of this study is the presence of detailed genetic data. Favorable genetics were more frequent in the young patients group than in the e-AML + s-AML group. S-AML patients carried the *KMT2A-AF9* gene more frequently than young patients with primary AML. Furthermore, s-AML patients who carried the mutation in this gene had a higher blast count in bone marrow than s-AML patients with no aberrations in this gene [14]. *KMT2A* is located on chromosome 11 and encodes for the transcriptional coactivator that plays a crucial role in regulating gene expression during early development and hematopoiesis [58]. S-AML patients had lower complete remission rates than young patients (s-AML vs. young: 58% vs. 83 %, *p* < 0.001). MRD was more frequent in s-AML patients than in young patients (*p* = 0.039). s-AML was a significant risk factor for the overall survival but had no effect on event-free survival [14]. 

## 7. Treatment Options in Secondary AML

### 7.1. Chemotherapy

Although the selection of therapy in AML is quite vast, the therapeutic options for patients with s-AML, for which efficiency has been confirmed in clinical trials, are limited. The historical standard for intensive induction in s-AML was “7 + 3” regimen, standing for 7 days of cytarabine continuous infusion, followed by 3 days of treatment with anthracycline [73]. However, this classical approach did not provide satisfactory responses: the overall survival, as well as the achievement of complete remission, was inferior in s-AML, compared to de novo AML [74]. The chemoresistance in s-AML is thought to be a consequence of numerous factors. As has been mentioned before, s-AML cells often harbor adverse-risked aberrations. Some of them lead to the generation of multidrug-resistant phenotypes. For example, P-glycoprotein, encoded by *MDR1*, is expressed on higher levels in patients with s-AML than in MDS and de novo AML patients. P-glycoprotein takes part in daunorubicin efflux. Intracellular daunorubicin levels are decreased as a result of efflux and this fact is independently associated with inferior overall survival. Chemoresistance is also a consequence of the upregulation of the antiapoptotic proteins, such as Bcl-2. Correspondingly, s-AML patients are often less fit due to prior malignancy and therapies. The previous treatment may result in the appearance of the chemoresistant clone [8,10]. Walter et al. [16] expressed the opinion that as a dominant s-AML clone derives from the MDS founding clone, the treatment strategy that allowed for the elimination of disease-propagating cells would be the one that targets the early mutations. 

For over twenty years, the treatment of s-AML was not different than the management of de novo AML, despite the fact that the effect of chemotherapy was significantly poorer. The introduction of CPX-351, which is liposomal daunorubicin and cytarabine in proportions of a 1:5 molar ratio, was a game changer. The liposomal carrier maintains the synergistic drug ration for over 24 h, leading to longer drug exposure [73]. Also, the usage of liposomes may potentially lead to bypassing P-glycoprotein efflux pumps, which results in increased intracellular levels of carried chemotherapeutics [75]. In August 2017, the FDA approved CPX-351 as the treatment for adults with newly diagnosed t-AML and AML with myelodysplasia-related changes [76]. The approval was a consequence of the results of the phase 3 clinical study, which demonstrated the improved outcomes in patients treated with CPX-351 over daunorubicin and cytarabine (“7 + 3”). The overall survival in patients treated with CPX-351 compared with the patients treated with “7 + 3” scheme was significantly improved (9.07. vs 5.95 months, HR:0.7; 95% CI: 0.53–0.95). The study was conducted on the group of patients aged 60 to 75 years [77]. The phase 2 study of CPX-351 (NCT04269213) in patients aged 18–59 years of age is ongoing; participants for the study are still being recruited [78]. However, Przespolewski et al. [79] provided us with a not-so-optimistic insight into the efficiency of CPX-351 in 30 patients younger than 60 years old. 60% of patients had AML-MRC, 23% had t-AML and 20% had antecedent MDS. There were 19 patients with a complex karyotype and 4 patients with a normal karyotype. The most common aberration was *TP53* mutation (in 36% of patients) and *FLT3-ITD* mutation (in 14% of patients). Complete remission or partial remission was achieved in 46.7% of patients, whereas 51.7% were non-responders. Overall survival was 7 months, with approximately the same overall survival rate achieved in the younger group of patients with s-AML, treated with conventional chemotherapy. Both response rates and overall survival were inferior compared to the data reported in the phase 3 trial in patients aged from 60 to 75 years [79]. The phase I/II trial (NCT02642965) is ongoing in a pediatric population (children over 1 and no more than 21 years old), evaluating side effects and the best dose of CPX-351 when given with fludarabine phosphate, cytarabine and filgrastim in relapsed/refractory AML [80]. Cooper et al. [81] claim that the toxicities were manageable, and protocol therapy was effective. 1/6 patients experienced grade 3 decrease in ejection fraction. Some other (at least grade 3) adverse effects included were fever/neutropenia in 45% of patients, infection in 47% of patients and rash in 40% of patients. No deaths due to toxicity were observed. Complete remission was achieved in 54% of individuals. In March 2021, the FDA approved CPX-25 as a treatment of newly diagnosed t-AML or AML with myelodysplasia-related changes in children older than 1 year [82].

Patients with s-AML who are not fit enough to undergo intensive chemotherapy can benefit from treatment with the hypomethylating agents azacytidine or decitabine [10]. Those agents are currently approved for the treatment of MDS, CMML and AML [83]. Seymour et al. [84] proved that azacitidine improves overall survival in older patients with AML with myelodysplasia-related changes compared to conventional care regimens. Among patients with intermediate-risk cytogenetics, median overall survival with azacytidine was 16.4 months, and with CCR it was 8.9 months (hazard ratio 0.73 [95%CI 0.48, 1.10]). 87% of patients in each treatment arm experienced grade 3–4 adverse effects, of which the most frequent was thrombocytopenia in patients treated with conventional chemotherapy and febrile neutropenia in patients treated with azacytidine. Meers et al. [85] reported the data on the efficacy and safety of decitabine in AML. The study proved that the treatment with decitabine leads to better overall survival and progression-free survival [85]. 

### 7.2. Targeted Therapies

There are no targeted therapies specifically approved to treat s-AML. s-AML patients have been excluded from numerous trials of targeted agents. However, bearing in mind that the number of mutations is higher in s-AML, one may assume that numerous s-AML patients may benefit from targeted treatment. The most discussed and/or interesting approaches in this context are described below.

As it has been proved, *IDH1* and *I**DH2* mutations are one of the most common genetic aberrations in s-AML. Ivosidenib and enasidenib are small-molecule inhibitors of mutant *IDH1* and mutant *IDH2*, respectively. They have been approved for the treatment of adults with relapsed/recurrent AML with *IDH1* and *IDH2* mutations. In the case of treatment with ivosidenib, the complete remission or complete remission with partial hematologic recovery was 30.4% (95% CI: 22.5–39.3), including the rate of complete remission accounting for 21.6% (95% CI: 14.7–29.8). The efficiency of ivosidenib in r/r AML is promising; however, no data on the efficiency specifically in s-AML has been reported. During the treatment, adverse effects of grade 3 or higher were, most commonly, prolongation of the QT interval (in 7.8% of the patients), the *IDH* differentiation syndrome (in 3.9%), thrombocytopenia or a decrease in the platelet count (in 3.4%), anemia (in 2.2%) and leukocytosis (in 1.7%) [86]. Ivosidenib has been also approved in the treatment of naïve AML patients with *IDH1* mutated who are older than 75 years or have comorbidities that preclude the use of intensive induction chemotherapy [87]. The overall response rate in relapsed/recurrent AML patients treated with enasidenib was 40.3% (95% CI, 33.0–48.0%); the median response lasted for 5.8 months. Median overall survival in patients with relapsed/recurrent AML was 9.3 months (95% CI, 8.2–10.9 months). Grade 3 to 4 adverse events were hyperbilirubinemia (12%) and *IDH*-inhibitor-associated differentiation syndrome (7%) [88]. *IDH* differentiation syndrome is a common and serious reaction occurring in about 40% of patients treated with ivosidenib or enasidenib. In Norsworthy et al.’s study [89], median time to onset was 20 days (ranging 1 to 78 days) and 19 days (raging 1 to 86 days) in patients treated with ivosidenib or enasidenib, respectively. Differentiation syndrome was first described in 1992 by Frankel et al. [90] as the adverse reaction, occurring during the treatment of acute promyelocytic leukemia with retinoids. The key symptoms are fever and dyspnea with no signs of infection. Other symptoms include peripheral oedema, weight gain, episodic hypotension, renal insufficiency and hyperbilirubinemia. The most common treatment is dexamethasone, applied at the earliest symptom of differentiation syndrome due to its possible life-threatening nature [91].

Another molecule that is more frequently mutated in s-AML than in de novo AML is *FLT3*. Internal tandem duplications in *FLT3*-*ITD* can be managed with *FLT3* inhibitors. Two of the *FLT3*-*ITD* inhibitors—midostaurin and gilteritinib—were approved by the FDA in relapsed/recurrent *FLT-3* mutated AML in April 2017 and November 2018, respectively [92,93]. The approval of midostaurin followed the promising results of phase 3 RATIFY trial [94], conducted in 18–59 year old patients with de novo AML, harboring mutations in *FLT-3.* The overall survival was longer in the midostaurin group than in the group treated with chemotherapy (HR: 0.78; 95% CI: 0.63 to 0.96; *p* = 0.009). Severe adverse events were similarly frequent in the two groups, although anemia and rash grade 3 and higher were more frequent in patients treated with midostaurin (92.7% vs. 87.8%, *p* = 0.03 and 14.1% vs. 7.6%, *p* = 0.008, respectively). The approval of gilteritinib was based on the results of ADMIRAL trial [95]. The median overall survival was higher in the gilteritinib group than in the chemotherapy group (9.3 months vs. 5.6 months; HR: 0.64; 95% CI: 0.49 to 0.83; *p* < 0.001). 34% of patients in the gilteritinib group and 15.3% of patients in the chemotherapy group achieved complete remission with full or partial hematologic recovery. Adverse events of higher than grade 3 were less frequent in the gilteritinib group than in the chemotherapy group (19.34 events per patient-year in the gilteritinib group and 42.44 events per patient-year in the chemotherapy group). The most common serious adverse events during gilteritinib therapy were febrile neutropenia and an increase in liver enzymes (ALT, AST). The most common adverse effects that led to death in the group treated with gilteritinib were pneumonia (1.2% of patients), large intestine perforation (0.8% of patients) and septic shock (0.8% of patients).

Venetoclax (ABT-199) is a highly BCL-2-selective, orally available BH3-mimetic that is best known for its usage in the treatment of chronic lymphocytic leukemia [96,97]. However, it has also been approved in the treatment of AML in patients over 75 years old or younger, but is not fit for intensive regimens. Venetoclax is used in combinations with hypomethylating agent or low-dose cytarabine, as its activity as a single agent is limited [98,99]. Venetoclax with azacytidine or decitabine has proven to be equally effective in s-AML as in de novo AML. However, durable responses were rare. In patients with *Flt3*, *RAS* and *TP53* mutation, response rates were poorer (50%, 33% and 47%, respectively) [8]. Serious adverse events during the treatment with venetoclax include pyrexia, autoimmune hemolytic anemia, pneumonia and febrile neutropenia. Neutropenia, infection, anemia and thrombocytopenia are the most common grade 3–4 adverse events [100,101].

Table 2 summarizes the selected targeted approaches in the treatment of AML, including secondary AML.

### 7.3. Chimeric Antigen Receptor—T Cell Therapy

One of the most interesting novel approaches to s-AML management is through use of the chimeric antigen receptor—T cell therapy (CAR-T). Zhang et al. [102] successfully applied CAR-T for the treatment of s-AML in a 10-year-old girl who had a relapse of acute lymphocytic leukemia, followed by leukemia originating from the myeloid lineage which was genetically proved to be AML. C-type lectin-like molecule-1 (CLL-1), highly expressed in s-AML blasts, was the target for the therapy. At least 9 months remission has been achieved at the time of the case report submission. The adverse effect was a cytokine release syndrome of grade I/II [102]; this is a common adverse effect of CAR-T cell therapy, along with B cell aplasia, neurotoxicity and infections [103]. CRS is the consequence of the sudden activation of CAR-T cells and, consequently, the release of cytokines (including IL-6, IL-10, TNF-α, and IFN-γ). Symptoms of CRS range from fever and hypoxia to renal failure and, rarely, death. The management strategy includes tocilizumab (IL-6 receptor antagonist) and corticosteroids [104]. In this case, tocilizumab and steroids were not needed. CAR-T therapy was also successfully used in the 6-year-old girl with s-AML and the history of Fanconi anemia and juvenile myelomonocytic leukemia [105]. In this case, the authors reported the usage of CLL1-CD33 cells. Technically, during this process, T lymphocytes are isolated from the blood and genetically modified to express chimeric antigen receptors (CARs): one specific for the CD33 antigen and one specific for the C-type-lectin-like molecule-1 (CLL-1). CAR-T cells are then expanded and injected into the patient’ s body, where they specifically target and bind to CD33- and CLL1-expressing tumor cells, with their anti-CD33 CAR and their anti-CLL1 CAR, respectively [106]. Both CD33 and CLL1 are overexpressed on myeloid leukemia cells, inducing selective toxicity in tumor cells that express the CD33 antigen and the CLL1 antigen [107]. 

### 7.4. Allo-HCT

The potentially curative treatment in s-AML is allogeneic hematopoietic cell transplantation (HCT). Yoshizato et al. [108] enrolled 797 patients diagnosed with MDS at initial presentation who underwent bone marrow transplantation. 25% of those patients had s-AML. The progression to the s-AML has been proved to be one of the 12 variables that affected the prognosis. Other significant factors included mutations in 4 genes (*NRAS*, *TP53*, *CBL* and *CK*) and 8 clinical factors: performance status, age at HCT, hematopoietic stem cell transplantation–specific comorbidity index (HCT-CI), female donor to male, complete remission at transplant and time from initial diagnosis to transplant, as well as red blood cell transfusion history [108]. 

Nilsson et al. [109] used the Swedish AML Registry, comprising 3337 adult patients, to compare the outcome of allo-HCT within the group of patients with s-AML and de novo AML. Complete remission has been reached in 72% of patients with de novo AML, and in 60% and 45% in the t-AML and AHD-AML groups, respectively (*p* < 0.001 for both t-AML and AHD-AML vs de novo AML). 576 patients (22%) with de novo AML, 74 patients (17%) with AHD-AML, and 57 patients (20%) with t-AML have undergone allo-HCT. Among the patients who did not undergo HCT, no patient with previous myeloproliferative neoplasm survived. A 5-year overall survival rate in patients with antecedent MDS and t-AML, with no allo-HCT, was 2% and 4%, respectively. HCT proved to be superior to chemotherapy consolidation, as the 5-year overall survival rate and relapse-free survival in patients with secondary AML in the first complete remission (CR1) was 48% and 43% for patients undergoing HCT versus 20% and 21% for patients receiving chemotherapy consolidation, respectively. The prognostic factors predicting outcomes in patients with secondary AML who underwent HCT in CR1 were analyzed in 100 patients. Survival was favorable when the graft source was peripheral stem cells rather than bone marrow, mild cGVHD versus no cGVHD took place and aGVHD of grade 0–I vs grade II–IV was present. There was no difference in survival associated with recipient age and sex, type of secondary AML, cytogenetic risk, donor age, female donor to male recipient HCT, early or late transplantation period (1997–2004 vs. 2005–2014), HCT-CI score, myeloablative or nonmyeloablative conditioning or cytomegalovirus reactivation [12]. 

In Litzow et al.’s study [110], conducted in the younger cohort (the median age of patients: 40 years old), consisting of patients with treatment related AML and treatment related MDS, risk factors for HCT included age over 35 years, poor-risk cytogenetics, refractory disease and the lack of a well-matched donor. Two-thirds of the patients had a pretransplantional t-AML diagnosis, most of whom had a history of lymphoma, breast cancer or ALL, with the median time of diagnosis from prior disease ranging from less than a year to 28 years (median time: 4 years) [109].

The ESMO guidelines [110] for diagnosis, treatment and follow-up in AML adult patients, as well as American Society of Hematology guidelines for treating newly diagnosed AML in the elderly [111] do not describe the clear treatment algorithms that should be followed in s-AML specifically. This is likely due to the fact that the therapeutic options for patients with s-AML, for which efficiency has been confirmed in clinical trials, are very limited. Figure 2 presents the proposal of the treatment algorithm in s-AML, which is based on the review of the literature (especially NCCN Clinical Practice Guidelines in Oncology [112] and clinical practice. 

## 8. Conclusions

Secondary AML is a subgroup of AML with poorer treatment response and prognosis than de novo AML. In light of the rapid increase in the number of patients undergoing chemotherapy and the number of cancer survivals, the rise of incidence of s-AML, including t-AML, seems inevitable. It is hard to resist the impression that, for years, s-AML was considered to be an “ugly stepsister” of de novo AML. In numerous analysis and clinical trials, s-AML was often analyzed both with de novo AML or completely excluded, which results in limited data on the biology of this type of AML. Consequently, data on the successful management of s-AML is still limited. 

In this article we focused our attention on the s-AML antecedent to MDS and t-AML, but it needs to be noted that AML may also be antecedent to other hematological conditions such as myeloproliferative neoplasms, MDS/MPN overlap syndromes (chronic myelomonocytic leukemia, atypical chronic myeloid leukemia), Fanconi anemia, aplastic anemia or paroxysmal nocturnal hemoglobinuria. As the genetic landfill and biology of these diseases differ significantly, the management and prognosis of s-AML antecedent to those conditions are also distinct and should be addressed separately. In the coming years, efforts should be concentrated both on the improvement of the already approved strategies in the treatment of s-AML, but also on the development of new approaches. Better understanding of genetics and epigenetics of s-AML would allow us to determine precise biologic drivers leading to leukogenesis, aiding in targeted treatment. Further evaluation of the genetic landscape and the development of alternative therapeutic approaches in secondary AML remain issues of critical importance. 

## Figures and Tables

**Figure 1 jcm-11-04283-f001:**
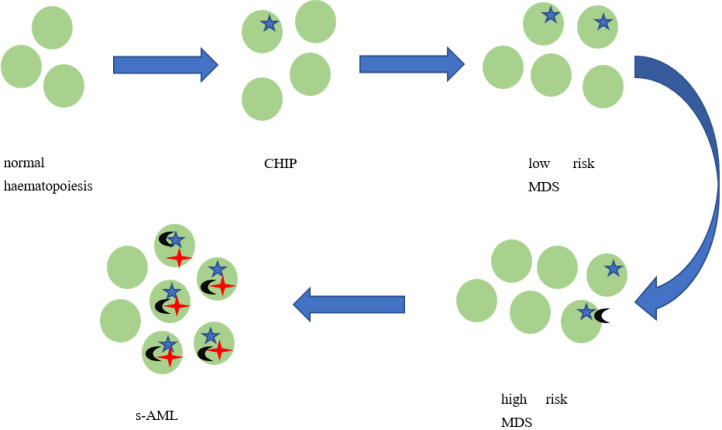
**Clonal evolution: from normal haematopoiesis to secondary AML.** MDS arises through the sequential acquisition of somatic mutations in recurrently involved genes. The earliest genetic step in MDS pathogenesis is “clonal haematopoiesis of undetermined potential” (CHIP)—An entity that describes the presence of somatic mutations in the blood or bone marrow cells, when no other criteria for hematologic neoplasia are met [19,20]. During MDS initiation, newly acquired mutations appear, while previous mutations are also carried forward. Some of those acquired mutations have no consequences as “passenger” mutations while others become driver mutations, contributing to clonal evolution [8]. The final stage of the disease progression is an evolution from MDS to s-AML, also occurring as a consequence of somatic mutations accumulation [18]. The whole process is called “clonal evolution” [8].

**Figure 2 jcm-11-04283-f002:**
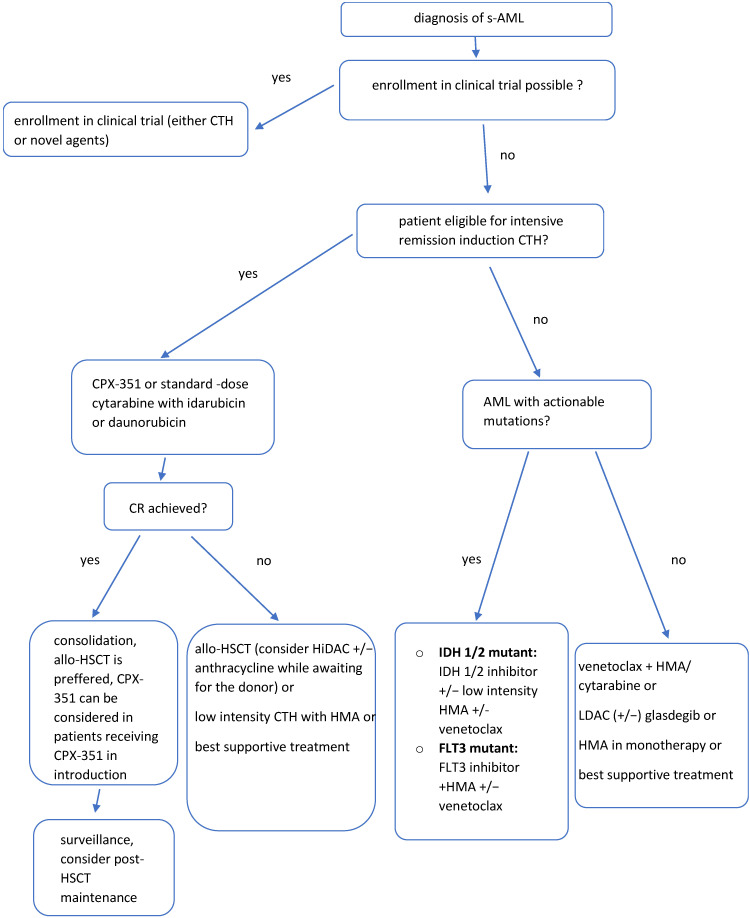
**The proposed treatment algorithm in s-AML [110,111,112,113].** Patients with s-AML generally have poor prognosis, that is why their enrollment in a clinical trial for treatment induction is strongly encouraged. The management depends on a patient’s eligibility for intensive remission induction. Allogeneic HSCT remains the best option for long-term disease control, so HLA testing should be performed promptly in those who may be candidates for either fully ablative or reduced-intensity conditioning. s-AML—secondary acute myeloid leukemia, CTH—chemotherapy, allo-HSCT—allogenic hematopoietic stem cell transplantation, HiDAC—high dose intermittent cytarabine, HMA—hypomethylating agents, LDAC—low-dose cytarabine.

**Table 1 jcm-11-04283-t001:** The mutations proved to be involved in leukemic transformation in patients with myelodysplastic syndrome (MDS) [8,27,43,44,45,46,47,48,49,50,51,52,53,54,55,56,57,58].

Mutated Gene	Cellular Location of Gene Products	Gene Product Function	the Risk of Leukemic Transformation in MDS	Discoverers of Gene Presence in MDS
*SRSF2*	nucleus	required for formation of splicing complex, interacts with spliceosomal components during spliceosome assembly	Mutations in *SRSF2* are associated with 4-fold increased risk of leukemic transformation.	Yoshida et al., 2011 [43]
*U2AF1*	nucleus	plays a critical role in RNA splicing by mediating interactions between the large subunit and proteins bound to the enhancers	Mutations in *U2AF1* are associated with 3-fold increased risk of leukemic transformation.	Graubert et al., 2011 [44]
*IDH1*	cytoplasm and peroxisomes	catalyzes the oxidative decarboxylation of isocitrate to 2-oxoglutarate, probably plays role in regeneration of NADPH for intraperoxisomal reductions	Mutations in *IDH1* are associated with 7-fold increased risk of leukemic transformation.	Kosmider, O. et. al. 2010 [45] Thol F. et al., 2010 [46]
*ASXL1*	nucleus	epigenic regulator involved in chromatin remodeling	Frameshift *ASXL1* mutations are associated with 2,4-fold increased risk of leukemic transformation.	Gelsi-Boyer V et al., 2009 [47]
*EZH2*	nucleus	epigenic regulator involved in chromatin remodeling	MDS patients with mutatons in *EZH2* have higher rate of transformation to sAML, however the precise hazard ratio for leukemic tranformation hasn’t been established yet.	Ernst et al., 2010 [48]
*NRAS*	cell membrane, Golgi apparatus membrane	binds GDP/GTP and possess intrinsic GTPase activity, when mutated and constitutively active it has oncogenic function	The relative risk of progression to s-AML is not established yet, patients with *NRAS* mutation have shorter time to transformation.	Hirai et al., 1987 [49]
*KRAS*	cytosol, plasma membrane, endomembrane system	binds GDP/GTP and possess intrinsic GTPase activity, plays an important role in the regulation of cell proliferation	The relative risk of progression to s-AML is not established yet, patients with *KRAS* mutation have shorter time to transformation.	Lyons J et al., 1988 [50]
*FLT3*	endoplasmic reticulum, nucleus	regulates differentiation, proliferation and survival of hematopoietic progenitor cells and dendritic cells	The relative risk of progression to s-AML is not established yet, however it was proved that patients with *FLT3* mutation have shorter time to transformation.	Horiike et al., 1997 [51]
*TP53*	nucleus, cytoplasm and cytosol, mitochondrion, cytoskeleton, endoplasmic reticulum	induces cell cycle arrest, apoptosis, senescence, DNA repair, or changes in metabolism in response to cellular stresses	MDS patients carrying *TP53* mutations present a higher frequency of karyotype abnormalities (for example -7 and complex karyotypes), which are associated with higher risk of leukemic transformation.	Jonveaux et al., 1991 [52]
*RUNX1*	nucleus	transcription factor involved in the development of normal hematopoiesis.	In MDS patients who progressed to s-AML, *RUNX1* mutations are present three times more frequently than in patients who didn’t experience progression [8].	Imai Y. et al. 2000 [53]
*DNMT3A*	cytoplasm and nucleus	involved in de novo DNA methylation, essential for the establishment of DNA methylation patterns during development	*DNMT3A* R882 mutant MDS cases were shown to have markedly increased risk of AML transformation (25.8%, vs. 1.7%).	Walter et al., 2011 [54]

**Table 2 jcm-11-04283-t002:** Selected targeted approaches in the treatment of AML including Secondary acute myeloid leukemia (AML) [86,87,88,94,95,99,101]. CR + CRh—complete remission/complete remission with partial hematologic recovery, ORR: overall response rate, r/r -relapsed/recurrent AML, OS—overall survival.

Targets	Drugs	Group of Patients	Clinical Benefit	Date of FDA Approval	Most Common Grade 3 or More Side Effects
** *IDH1* **	ivosidenib	adults with relapsed/recurrent AML with IDH1 mutations adults with newly-diagnosed AML with a susceptible IDH1 mutation, more than 75 years old or with comorbidities that preclude the use of intensive induction chemotherapy	CR + CRh in 30.4% of patients (95% CI: 22.5–39.3), including the CR: 21.6% (95% CI: 14.7–29.8) (NCT02074839, phase 1) [86] CR + CRh in 42.9% of patients (95% CI: 24.5, 62.8), 41.2% of the transfusion-dependent patients achieved transfusion independence for at least 8 weeks (NCT02074839, phase 1) [87]	July 2018 May 2019	prolongation of the QT interval, the IDH differentiation syndrome, thrombocytopenia or the decrease in the platelet count, anaemia, leucocytosis
** *IDH2* **	enasidenib	adult patients with r/r AML with *IDH2* mutations	ORR in r/r AML: 40.3% (95% CI, 33.0–48.0%), Median OS: 9.3 months (95% CI, 8.2–10.9 months) (NCT02577406, phase 3) [88]	August 2017	hyperbilirubinemia, IDH-inhibitor-associated differentiation syndrome
** *FLT3* **	midostaurin gilteritinib	adults with r/r AML with *FLT-3* mutated	OS longer in the midostaurin group than in the group treated with chemotherapy median OS: 31.5 months in the midostaurin group and 25.6 months in the placebo group (HR: 0.78; 95% CI: 0.63 to 0.96; *p* = 0.009) (phase 3 RATIFY trial, NCT00651261) [94] The median OS s higher in the gilteritinib group than that in the chemotherapy group (9.3 months vs. 5.6 months; HR: 0.64; 95% CI: 0.49 to 0.83; *p* < 0.001) (ADMIRAL trial, phase 3, NCT02421939) [95]	April 2017 November 2018	febrile neutropenia, infections, neutropenia, anemia, thrombocytopenia febrile neutropenia, elevated liver enzymes
** *BCL-2* **	venetoclax	in combination with azacitidine, decitabine, or low-dose cytarabine for newly-diagnosed AML in adults 75 years or older or who have comorbidities precluding intensive induction chemotherapy	median OS 14.7 months (95% CI: 11.9, 18.7) in patients treated with venetoclax plus azacitidine vs. 9.6 months (95% CI: 7.4, 12.7) in patients receiving placebo plus azacitidine (HR 0.66; 95% CI: 0.52, 0.85; *p* < 0.001); CR rate: 37% (95% CI: 31%, 43%) vs. 18% (95% CI: 12%, 25%) in venetoclax plus azacitidine vs. placebo plus azacytidine (VIALE-A, NCT02993523) [99] CR rate on the venetoclax plus LDAC: 27% (95% CI: 20%, 35%) vs. 7.4% (95% CI: 2.4%, 16%) in patients receiving placebo plus LDAC, but no significantly improved OS in patients LDAC plus venetoclax vs. placebo plus LDAC (HR 0.75; 95% CI 0.52, 1.07; *p* = 0.114), VIALE-C (NCT03069352) [101]	accelerated approval: November 2018. regular approval: October 2020	neutropenia, infection, anaemia, thrombocytopenia

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
