# Peer review of "In Pursuit of Genetic Prognostic Factors and Treatment Approaches in Secondary Acute Myeloid Leukemia—A Narrative Review of Current Knowledge"

_jcm, 2022, doi:10.3390/jcm11154283_

Round 1
Reviewer 1 Report
Dear Author,
the manuscript entitled " In pursuit of genetic prognostic factors and treatment approaches in Secondary Acute Myeloid Leukemia -a narrative review of current knowledge" contains a comprehensive description of the current knowledge on genetic prognostic factors and treatment of patients with secondary acute myeloid leukemia. The work is based on 91 items of literature, most of which come from recent years. Some of the items are older than 10 years, but their use is justified by reference to the source publications. The manuscript is divided into clearly defined and logical sections to facilitate its analysis and understanding
I believe that the manuscript of of a high quality and I only have a few minor comments
1. In the "introduction" chapter, data on the number of AML cases in Poland should be provided, possibly in Europe, not the USA.
2. Legends to the tables should be placed above the tables
Reviewer 2 Report
The manuscript entitled “In pursuit of genetic prognostic factors and treatment approaches in Secondary Acute Myeloid Leukemia -a narrative review of current knowledge “ by Paulina Stefaniuk et al. presents a concise review of s-AML and what makes it a distinct disease from AML at the genetic and molecular level. It is very interesting and presents a great overview of the literature. I have a few major comments to clarify a few sentences in the manuscript and to edit Table 1.
Major comments
1. In table 1, it would be great to replace the column “the significance in leukemic transformation” to a column that describes the role of each gene. Even though all of those genes might share a similar function, it would be useful for the reader to know what exactly this genes does, what kind of risk is linked to the gene, and to cite the studies that found the mutations with the last authors name. It would be like an overview of the text, and it is a great way to give credit to the people who discovered the mutations in MDS patients.
2. On line 316: “The most frequently, malignancies that later result in t-AML are breast cancer and lymphomas (8)”. I don’t understand this sentence. Do the authors mean that the most frequent malignancies that result in t-AML are breast cancers and lymphomas? It needs rephrasing perhaps.
3. On line 366: “This data proves that t-AML is in fact a heterogeneous group, that deserves 367 to be furtherly evaluated in clinical trials and approached in a personalized manner” “Prove” might be a strong word in this context. I would change “prove” to “highlight”.
4. On line 385: “No difference has been observed in the number of patients with poor cytogenetics when comparing MDS-AML and MPN-AML.” Is the number a frequency, or an absolute number?
5. On line 464: “The study proved that decitabine leads to better overall survival and progression free survival in real-life settings”. This paragraph talks about “real-life settings”. What do the authors mean by this exactly? Aren't all clinical trials are done in real-life settings no? I am not sure I understand.
6. On line 549: “The second case of successful usage of CAR-T therapy in s-AML reported the usage of CLL1-CD33 cells in 6-year-old girl with previous Fanconi anemia and juvenile myelomonocytic leukemia (88).” It would be important to describe the CAR-T cell design (CCL1-CD33 compound CAR-T cells) because it is something that is a bit newer, so some readers might not know about this technology.
Minor comments
1. On line 37: specify that it is 12,2/100 000 cases.
2. On line 43: “what allow to diagnose”. “What” should be replaced with “which”.
3. One line 104: “CHIP is a significant, independent prognostic factor of hematologic malignancies in the future”. This phrase might need to be rewritten since it sounds like you are talking about the future, but this has already been implemented?
4. One line 236: “RUNX1 transcription factor regulates both normal and abnormal hematopoiesis, the lack of RUNX1 activity leads to hematopoiesis defects and is embryonic lethal”. Please insert the reference.
5. On line 257: “What is interesting, cohesin mutations occurred in more than the half of the patients with acute megakaryocytic leukemia associated with Down syndrome (41)” This sentence needs to be reformatted.
6. One line 260: “It is not a coincidence that they are described at the end of the review’s part raising the subject of mutations leading to the development of s-AML.” Is this sentence necessary? I feel like it might break the flow of the manuscript.
7. On line 302: “To be precise, it needs to be mentioned, that Revised 4th edition of the WHO classification uses the term t-MN (therapy-related myeloid 304 neoplasia), rather than t-AML.” This sentence needs to be reformatted. I would suggest to remove “it needs to be mentioned”.
8. On line 326: “What is interesting is the fact that TP53 mutations are present not only before the onset of the t-AML, but also before chemotherapy exposure!” I would suggest to remove the exclamation mark and replace it by a dot.
9. On line 350: “What is interesting in patients with BRCA1 and BRCA2 mutated, the latency time was significantly longer.” This sentence is grammatically incorrect.
10. On line 390: “To conclude, AHD-AML was an independent poor prognostic factor, but in case of t-AML it wasn’t proved (9).” This sentence is grammatically incorrect.
11. On line 395: “KMT2A-AF9 gene was the most frequently mutated in s-AML patients, also s-AML patients carrying mutation in this gene, had higher blast count in bone marrow (14).” This sentence is grammatically incorrect.
12. On line 413: “This glycoprotein takes part in daunorubicin efflux, what results in decreased intracellular levels of this chemotherapeutic, thus poorer response. In s-AML antiapoptotic proteins, such as Bcl-2, are overexpressed. Chemoresistance is also contributed to the fact, that s-AML patients are often less fit as a consequence of prior malignancy and therapies.” All three sentences are grammatically incorrect.
13. On line 422: “The liposomal carrier enables to maintain the synergistic drug ration for over 24 hours, what allows enables to overcome the pharmacokinetic differences between daunorubicin and cytarabine (56)”. This sentence is grammatically incorrect.
14. On line 453: “Seymour et al. (67) proved that in AML with myelodysplasia related changes overall survival is better in patients treated with azacytidine vs conventional regimens (8.9 months vs 4.9 months; HR: 0.74, 95%CI 0.57- 0.97).” This sentence is grammatically incorrect.
15. On line 607: “Authors’ opinion the further evaluation of genetic risk factors for leukemic transformation in different hematological malignancies and the development of alternative therapeutic approaches in s-AML remain the issues of critical importance.” This sentence is grammatically incorrect.
Reviewer 3 Report
In this manuscript, Stefaniuk et al. review the search for novel genetic predictors and biomarkers for secondary acute myeloid leukemia (AML). The text is well written and informative, although a few issues require further care, as outlined below.
MAJOR ISSUES
1. Lines 39-40: the authors righteously state that the outcome of AML has improved steadily from the ‘70s to today. However, the authors should clarify in more detail the relative contribution of improved supportive care, reduced transplant mortality and availability of novel medicines.
2. The section entitled “Mutations leading to leukemic transformation” should be less descriptive and more interpretative. In particular, the authors should clarify that some mutations are actionable and may represent the target for novel molecular medicines.
3. A figure depicting the cellular location/function of the gene products of the genes mutated in secondary AML would be desirable and useful to the readers.
4. A graphic scheme representing the algorithm for treatment of secondary AML should be included and would be of great help to the reader.
5. The authors fail to mention atypical chronic myeloid leukemia as a disease that may evolve to secondary AML. The authors should include a statement on this topic, also referring to a recent review on this condition (Crisà E, et al. Atypical Chronic Myeloid Leukemia: Where Are We Now? Int J Mol Sci. 2020;21(18):6862).
6. The authors should include a short paragraph on quality of life (QoL), also referring to important recent data in this field (Gamper EM, et al. The EORTC QLU-C10D was more efficient in detecting clinical known group differences in myelodysplastic syndromes than the EQ-5D-3L. J Clin Epidemiol. 2021;137:31-44).
MINOR ISSUES
1. Line 62: “scare” should be “scarce” or “scant”
2. The English language requires minor revisions throughout the text
3. The manuscript may benefit from a certain reduction in length (10% reduction)
Round 2
Reviewer 2 Report
All my comments have been answered. Thank you!
Reviewer 3 Report
The authors have adequately addressed all the issues that had been raised. No further comments from my side.